# Characterizing the Effect of *Campylobacter jejuni* Challenge on Growth Performance, Cecal Microbiota, and Cecal Short-Chain Fatty Acid Concentrations in Broilers

**DOI:** 10.3390/ani14030473

**Published:** 2024-01-31

**Authors:** Walid G. Al Hakeem, Emily E. Cason, Daniel Adams, Shahna Fathima, Revathi Shanmugasundaram, Jeferson Lourenco, Ramesh K. Selvaraj

**Affiliations:** 1Department of Poultry Science, The University of Georgia, Athens, GA 3060, USA; walid.alhakeem@uga.edu (W.G.A.H.); eec32271@uga.edu (E.E.C.); daniel.adams1@uga.edu (D.A.); shahna.fathima@uga.edu (S.F.); 2Toxicology and Mycotoxin Research Unit, US National Poultry Research Center, Athens, GA 30605, USA; revathi.shan@usda.gov; 3Department of Animal and Dairy Science, The University of Georgia, Athens, GA 30602, USA

**Keywords:** *Campylobacter jejuni*, gut microbiota, broiler

## Abstract

**Simple Summary:**

*Campylobacter jejuni* (*C. jejuni*) is the most common cause of bacterial gastroenteritis in humans worldwide. Poultry and poultry products serve as major reservoirs for this bacterium. In poultry, *C. jejuni* colonizes the ceca of broilers with a high load without compromising the bird’s growth. Understanding the interaction between the host and the microbe is essential for the development of innovative strategies to control *C. jejuni* in poultry. The objective of this study was to understand the impact of *C. jejuni* on bird growth, cecal microbiota, and cecal short-chain fatty acid concentration. Throughout our study, *C. jejuni* did not impact the bird’s performance. However, the *C. jejuni* challenge led to a decrease in the number of observed bacteria compared to the control group. At the species level, the *C. jejuni* challenge decreased the relative abundance of beneficial bacteria, such as *Sellimonas intestinalis*, and increased the relative abundance of *Faecalibacterium sp002160895* compared to the control group. Despite the changes in the microbial composition, the *C. jejuni* challenge did not change the microbial function or the concentration of short fatty acids in the ceca compared to the control group. In conclusion, *C. jejuni* infection in broilers can alter the microbial composition without compromising the bird’s performance.

**Abstract:**

This study aimed to understand the effect of *C. jejuni* challenge on the cecal microbiota and short-chain fatty acid (SCFA) concentration to form a better understanding of the host–pathogen interaction. Sixty broilers were randomly allocated into two treatments: control and challenge. Each treatment was replicated in six pens with five birds per pen. On day 21, birds in the challenge group were orally gavaged with 1 × 10^8^
*C. jejuni/*mL, while the control group was mock challenged with PBS. The *C. jejuni* challenge had no effect on body weight, feed intake, and feed conversion ratio compared to the control group. On day 28, the *C. jejuni* challenge decreased the observed features and Shannon index compared to the control group. On the species level, the *C. jejuni* challenge decreased (*p* = 0.02) the relative abundance of *Sellimonas intestinalis* on day 28 and increased (*p* = 0.04) the relative abundance of *Faecalibacterium sp002160895* on day 35 compared to the control group. The *C. jejuni* challenge did not change the microbial function and the cecal concentrations of SCFA on days 28 and 35 compared to the control group. In conclusion, *C. jejuni* might alter the gut microbiota’s composition and diversity without significantly compromising broilers’ growth.

## 1. Introduction

*C. jejuni* is the leading cause of gastrointestinal disease in the United States [1,2]. The financial impact attributed to *C. jejuni* infection is around USD 2 billion [3]. *C. jejuni* is highly prevalent among poultry species, including broiler breeders, broilers, turkeys, and ducks [4,5]. Among these, broilers are the main reservoir for *C. jejuni*, establishing a persistent and benign infection that can reach up to 1 × 10^9^ CFU/g in the ceca [6]. The majority of *C. jejuni* cases are associated with the consumption of contaminated poultry products [7]. Typically, broilers are infected with *C. jejuni* at 2–3 weeks of age, and *C. jejuni* persists until slaughter day [8]. During slaughter, the release of cecal contents may result in cross-contamination, ultimately producing contaminated poultry products [9,10].

In humans, *C. jejuni* infection manifests as a self-limiting enteric disease characterized initially by fever and vomiting, followed by abdominal pain that transitions into watery or bloody diarrhea [11,12]. Conversely, in broilers, *C. jejuni* is a near-commensal bacterium, leading to asymptomatic infection [13]. The mechanism behind this dichotomy between benign infection in broilers and clinical disease in humans remains poorly understood. Efforts to elucidate these differences involved investigations into the role of avian intestinal mucus in conferring protection against *C. jejuni* [14,15,16]. The difference in structure and avian mucus efficacy in trapping pathogens is a major reason behind the benign infection in broilers [16,17]. Additionally, different reports emphasized the commensal nature of *C. jejuni* infection in broilers, highlighting the absence of an effective immune response despite the high colonization rate in the ceca [13,18]. Nonetheless, other papers have challenged the notion of *C. jejuni* being merely a commensal bacterium, citing observed reduction in performance parameters following the infection of *C. jejuni* [19,20,21,22].

The progression in DNA-sequencing techniques has enhanced our comprehensive understanding of *C. jejuni*’s enteric lifestyle within broilers [23]. Profiling the gut microbiota through sequencing has revealed the impact of *C. jejuni* infection on the microbial composition in the ceca, manifesting as an increase in the microbial diversity and increasing the relative abundance of *Firmicutes* at the expense of *Bacteroidetes* [24]. At the same time, other studies have reported an increase in the relative abundance of *Clostridium* [25], *Streptococcus*, and *Blautia* [24] compared to the non-infected groups.

Previous studies have primarily focused on sequencing a small fragment of the 16S rRNA gene, typically the V3-V4 regions. However, in the current study, a more comprehensive approach was utilized, which consisted of sequencing the entire 16S rRNA gene (V1–V9 regions) [26,27]. This approach should provide a more detailed result concerning the microbial composition. We hypothesized that *C. jejuni* would alter the gut microbial communities and function, leading to impaired bird growth. Therefore, this study was carried out to evaluate the effects of a *C. jejuni* challenge on performance production, cecal microbial composition, expression of microbial functional pathways, and SCFA concentrations in broilers.

## 2. Materials and Methods

All animal protocols utilized in this experiment were approved by the Institutional Animal Care and Use Committee (IACUC) at the University of Georgia (AUP: A2021 6-012-Y1-A2).

### 2.1. Birds and Experimental Setup

The 35-day experiment was conducted with 60 one-day-old Cobb 500 broilers, randomly allocated into two treatments: control and challenge. Each treatment was replicated in six pens with five birds per pen. The birds had ad libitum access to water and feed, and the broilers were fed a basal diet (as shown in Table 1) throughout the experiment. On day 21, the birds in the challenge group were orally gavaged with 1 × 10^8^
*C. jejuni*/mL, while the control group was mock challenged with PBS [13]. Body weight and feed intake were recorded weekly. The average feed intake and body weight gain were adjusted for mortality to calculate the feed conversion ratio per pen.

### 2.2. C. jejuni Challenge Preparation

Under microaerobic conditions, the *C. jejuni* ATCC 33650 strain was cultivated on a campy-CEFEX agar and incubated at 42 °C for 48 h. After incubation, the bacterial cells were collected using a sterile inoculum loop and suspended in PBS (NaCl, Sigma Chemical Co., St. Louis, MO, USA). Subsequently, the optical density of the bacterial suspension was adjusted at an OD value of 0.2, as measured at 540 nm using a spec-20. The measured OD value of 0.2 corresponded to 1 × 10^8^ CFU/mL of *C. jejuni* [28]. This equivalence was further confirmed through serial dilution and plating on Campy-CEFEX plates.

### 2.3. Cecal Sample Collection and DNA Extraction

One bird per pen was euthanized on days 28 and 35 of the experiment, and cecal samples were collected under aseptic conditions into a sterile cryogenic tube. The samples were immediately frozen using liquid nitrogen. DNA extraction was performed using a hybrid protocol [29]. This protocol combines enzymatic and mechanical methods to optimize DNA extraction from the cecal content. Briefly, 0.35 g of the cecal sample was transferred into a 2 mL lysing matrix E tube (MP Biomedicals LLC, Irvine, CA, USA). The mechanical disruption of cecal samples was carried out using a QIAGEN vortex adapter for the Vortex-Genie 2 vortex (QIAGEN, Venlo, The Netherlands) for 10 min at maximum speed. A QIAamp Fast DNA Stool Mini Kit (QIAGEN, Venlo, The Netherlands) was used for the enzymatic extraction. Upon completing the DNA extraction, the concentration and the purity of the DNA were checked via spectrophotometry using a Synergy H4 Hybrid Multi-Mode Microplate Reader along with the Take3 Micro-Volume Plate (BioTek Instruments Inc., Winooski, VT, USA). Samples with concentrations lower than ten ng/µL were disqualified, and the DNA extraction process was repeated.

### 2.4. DNA Sequencing and Bioinformatics Analysis

Following the DNA extraction, all nine variable regions of the 16S rRNA gene were sequenced (regions V1 to V9), as described earlier [30], at Loop Genomics (San Diego, CA, USA). The whole 16S rRNA gene was synthetically reconstructed from a series of standard Illumina PE150 reads, which were assembled to rebuild the whole 16S rRNA gene, as established previously [26].

The resulting DNA sequences were converted to FASTQ files and were imported into QIIME 2 [31]. The control of sequence quality and filtration of chimeras were achieved using the QIME 2 DADA2 plugin [32]. An amplicon sequence variant (ASV) frequency table and a summary of each ASV length were generated. A phylogenetic tree was generated using the QIME 2 phylogeny plugin [33]. Taxonomic classification was performed using the QIME 2 feature-classifier plugin, which uses the Naïve Bayes classifier trained on the SILVA 138 SSU database [34]; the reads were classified via taxon using the fitted classifier [35]. The ASV table was rarefied to a common sampling depth of 1331 sequences/samples for alpha and beta diversity analyses. The following alpha diversity indices were computed: number of observed features (ASV), Shannon diversity index, Faith’s phylogenetic diversity index, and Pielou’s evenness index. For beta diversity, Bray–Curtis distances were used. The relative bacterial abundance was also quantified at the phylum, family, genus, and species levels. Finally, the Phylogenetic Investigation of Communities by Reconstruction of Unobserved States (PICRUSt2) was used to make inferences about the metabolic functions of the microbial community [36], and metagenome metabolic functions were assessed using the MetaCyc pathway database.

### 2.5. Short-Chain Fatty Acid Analysis

Short-chain fatty acid analysis was carried out as described previously [37]. Briefly, the cecal contents were collected on days 28 and 35 of the study and stored in a cryogenic tube. The samples were immediately frozen using liquid nitrogen and stored at −80 °C until further analysis. On the day of analysis, 1 g of each sample was diluted in 3 mL of distilled water and placed into 15 mL conical tubes. The samples were then homogenized via vortexing for 30 s, and 1.6 mL of each sample was transferred into a new tube and centrifuged at 10,000× *g* for 10 min. Following, 1 mL of supernatant was transferred to a new tube and mixed with 200 µL of a metaphosphoric acid solution (25% *w*/*v*). Each sample was vortexed for 30 s to ensure proper mixture and stored at −20 °C overnight. The next day, the samples were thawed and centrifuged at 10,000× *g* for 10 min. The supernatant was then placed into polypropylene tubes with ethyl acetate in a 2:1 ratio of ethyl acetate to the supernatant. The samples were vortexed for 10 s and then left to settle for 5 min. Next, 600 µL of the top layer was transferred into screw-thread vials to analyze the SCFA concentrations. A Shimadzu GC-2010 Plus gas chromatograph (Shimadzu Corporation, Kyoto, Japan) with a flame ionization detector and a capillary column (Zebron ZB-FFAP; 30 m × 0.32 mm × 0.25 µm; Phenomenex Inx., Torrance, CA, USA) was used for SCFA analysis. This equipment utilized helium as the carrier gas. The sample injection volume was 1.0 µL. The column temperature started at 110 °C and gradually increased to 200 °C. The injector and detector temperatures were set to 250 °C and 350 °C, respectively. The samples’ peak heights were compared to standards to determine the concentrations of SCFAs in the samples.

### 2.6. Statistical Analysis

A *t*-test was used to analyze the effect of *C. jejuni* on the performance production and concentrations of SCFAs, with the pen being considered the experimental unit. The individual microbial taxa and alpha diversity indices were analyzed using the Kruskal–Wallis H test. Differences in beta diversity were assessed via permutational multivariate analysis of variance (MANOVA—Adonis). The *p*-values were statistically significant when *p* ≤ 0.01 and considered a trend between 0.01 and 0.05.

## 3. Results and Discussion

### 3.1. Performance Parameters

Broilers serve as the principal reservoir for thermotolerant *Campylobacter* species, particularly *C. jejuni*, which establish colonization within the avian gastrointestinal tract between two and three weeks of age [38]. The considerable prevalence of *C. jejuni* in broilers implies its ability to outcompete different commensal bacteria, establishing a persistent infection that persists until slaughter day [39]. *C. jejuni* is usually detected in broilers around two to four weeks of age and is rarely detected at younger ages [38]. The absence of *C. jejuni* in newly hatched chicks can be partially attributed to maternal antibodies [40]. In our experiment, broilers were challenged with *C. jejuni* at three weeks of age to mimic the spread of *C. jejuni* infection within poultry farms. The *C. jejuni* challenge did not affect the performance production, feed intake, and feed conversion ratio (Table 2) compared to the control group (*p* > 0.05). These findings agree with results reported in different studies, as the *C. jejuni* challenge on day 14 [13] or day 21 [18] did not alter the performance parameters compared to the control group (*p* > 0.05). Such results highlight the near-commensal nature of *C. jejuni* and its relatively harmless infection, whose effects are typically limited to the gut. However, it should be noted that other studies have reported a decrease in body weight gain in *C. jejuni*-challenged broilers on days 35 [20] and 49 [41] of age. In these studies, the difference in production performance was observed four weeks post-inoculation, a time not covered in our study.

### 3.2. Microbial Diversity

After completing all filtering steps, the resulting number of sequences across the samples ranged from 1331 to 11,268 and were rarified to a sequencing depth of 1331 sequences per sample for the calculation of diversity indexes. A total of 7 phyla, 99 genera, and 83 species were identified in the ceca, in addition to 35% of other species that were unclassified by the used classifier. Table 3 summarizes the different computed alpha diversity indices. The gut microbiota play a central role in mediating colonization resistance against foreign microorganisms [42]. However, microorganisms have developed a set of virulence factors that enable them to bypass colonization resistance and persist in the gut [43]. In our study, *C. jejuni* decreased (*p* ≤ 0.05) alpha diversity, namely, the number of observed features and the Shannon diversity index, compared to the control group on day 28. The decrease in these alpha diversity indices reflects a reduction in microbial diversity and richness compared to the control group. Our results align with previously reported results as the *C. jejuni* challenge decreased the microbial diversity in the ceca compared to the control group [23]. Broilers were infected with *C. jejuni* at day 21 of age, and it is possible that *C. jejuni* established a persistent colonization load that led to a decrease in diversity as it dominated the ceca. However, the microbial diversity in the ceca was restored at day 35 of age, as no difference was recorded between the control and the *C. jejuni*-challenged birds. *C. jejuni’*s ability to persist in infection without causing a chronic shift in the microbial communities highlights its near-commensal nature. The beta diversity (Figure 1) between the samples was calculated via the Bray–Curtis dissimilarity, which showed a clear separation between the control and challenged groups (*p* < 0.01). The differences observed in the alpha diversity indices were mirrored in the beta diversity, indicating a possible shift in the overall structure of the microbial communities following *C. jejuni* challenge. It was also noted that *C. jejuni-*infected broilers samples were not as clustered as the control group samples, indicating a higher variability within *C. jejuni-*infected broilers compared to the control group.

### 3.3. Bacterial Diversity at the Phylum Level

Table 4 shows the relative abundance of the phyla on days 28 and 35, in which *Firmicutes* and *Bacteroidota* dominated the cecal microbiota, constituting more than 93% of the total classified phyla in the ceca. These results align with previously reported numbers, as *Firmicutes* dominated the cecal, ileal, and jejunal microbiota found in broilers [42]. Concerning the changes in gut microbial composition, on day 28, the *C. jejuni* challenge increased the relative abundance of *Cyanobacteria* compared to the control group. *Cyanobacteria* constitute a diverse bacterial phylum that produces an extensive array of toxins known as cyanotoxins [44]. The exposure of birds and other animals to cyanotoxins, originating from blooms of aquatic *Cyanobacteria*, is well documented as a cause of neurologic and hepatic disease [45]. However, the role of *Cyanobacteria* in broiler gut microbiota remains insufficiently understood. A study has hinted at a potential correlation between the increased relative abundance of *Cyanobacteria* and increased production of IL-10 in broilers [46]. Notably, *C. jejuni* is considered a near-commensal organism within broilers. Establishing such commensal colonization necessitates a swift regulation of the pro-inflammatory immune response [47]. The increase in the relative abundance of *Cyanobacteria*, subsequently leading to an increased production of IL-10, can partially elucidate the tolerogenic immune response exhibited by broilers [46]. This response facilitates *C. jejuni’s* asymptomatic colonization and aids its persistence until the slaughter age.

Similarly, the relative abundance of *Desulfobacterota* increased following *C. jejuni* challenge compared to the control group. *Desulfobacterota* is a phylum mainly present in marine environments and the mammalian intestinal tract; it consists of many sulfur-reducing bacteria [48]. The main sources of sulfur in the diet are cysteine and methionine, while the main source of sulfur supplied by the host is the sulfated mucins [49]. In our experiment, both groups received the same diet. Hence, the dietary source cannot explain the increase in *Desulfobacterota*. One possible explanation is that *C. jejuni* infection increases the presence of mucin-degrading bacteria that can liberate L-fucose, which *C. jejuni* can use [50]. The degradation of mucus results in the liberation of sulfur, thus increasing the relative abundance of *Desulfobacterota*. Future experiments should investigate different metals such as iron, copper, and sulfur, which play a vital role in bacterial metabolism and function, to gain a broader understanding of the several changes occurring in the gut.

### 3.4. Bacterial Diversity at the Genera and Species Level

The top 10 most abundant genera are shown in Figure 2, and the significantly different genera on days 28 and 35 are shown in Table 5. Similarly, the 10 most abundant species are illustrated in Figure 3, and the significantly different species on days 28 and 35 are presented in Table 6. The cecal microbiota were mainly composed of *Alistipes_A_871400, Romboutsia_B*, and *Faecalibacterium* on day 28, while *Alistipes_A_871400*, *Romboutsia_B*, and *Gemmiger_A_73129* dominated the cecal microbiota on day 35. These genera were identified at the species level, with *Alistipes excrementavium*, *Romboutsia ilealis*, and *Faecalibacterium sp002160895* dominating the cecal microbiota on days 28 and 35. *Alistipes excrementavium* and *Romboutsia ilealis* are newly characterized bacteria that were recently detected in the ceca of broilers [51], explaining the lack of studies concerning the role of both species in broilers. On the other hand, *Faecalibacterium sp002160895* belongs to the *Faecalibacterium* genera, widely recognized as beneficial inhabitants of the gut ecosystem, exerting a range of beneficial effects such as actively competing with pathogenic bacteria for nutrients and adhesion sites [52].

Interestingly, in our experiment, the *C. jejuni* challenge increased the relative abundance of *Faecalibacterium sp002160895* compared to the control group. This result aligns with previous experiments, as the *C. jejuni* challenge also led to an increase in *Faecalibacterium*’s abundance [23]. The changes in *Faecalibacterium sp002160895* did not have an impact on the bird’s performance as the performance remained similar in both groups (*p* > 0.05). *Faecalibacterium* spp. are well-characterized producers of bioactive anti-inflammatory molecules such as shikimic and salicylic acids [53]. These anti-inflammatory molecules regulate the NF-κB/MAPK signaling pathway to reduce inflammation in the gut [53]. As previously mentioned, *C. jejuni* infection results in a tolerogenic immune response in broilers, and it appears that the increased relative abundance of *Faecalibacterium sp002160895* enhances such a response through the production of anti-inflammatory compounds.

The *C. jejuni* challenge decreased the relative abundance of several genera, such as *Sellimonas* on day 28 and *Acutalibacter* and *Oribacterium* on days 28 and 35, compared to the control group. *Acutalibacter* is a genus of bacteria from the family of *Acutalibacteraceae* with one known species (*Acutalibacter muris*) [51]. Similarly, *Oribacterium* is a genus of bacteria from the family of *Lachnospiraceae* [51]. However, little information is known concerning the role of *Acutalibacter* and *Oribacterium* in the broiler gut microbiome.

*Sellimonas* genera were classified on the species levels as *Sellimonas intestinalis*. *C. jejuni* challenge decreased the relative abundance of *Sellimonas intestinalis* compared to the control group. Little information is presented regarding the role of *Sellimonas intestinalis* in broilers. However, *Sellimonas intestinalis* was suggested as a potential biomarker for gut hemostasis, as its relative abundance increased in several patients recovering from different dysbiosis events [54]. The decrease in the relative abundance of *Sellimonas intestinalis* on day 28 could potentially help *C. jejuni* in colonizing the ceca.

Despite the changes in the microbial composition and the decrease in the gut diversity following *C. jejuni* challenge, the predicted microbial function shown in Figure 4 remained unchanged compared to the control group. Similarly, the *C. jejuni* challenge did not affect the production of SCFA in the ceca shown in Table 7. The SCFAs are important in maintaining gut integrity and function. A higher concentration of SCFA compared to the small intestine activates acetogenesis-dependent genes in *C. jejuni*, enabling the bacteria to adhere and colonize the ceca [55]. Furthermore, acetate is considered a major energy carbon/energy source for *C. jejuni*. Although only numerically different in the current study, a high concentration of acetate in the ceca can support *C. jejuni’s* high load and allow *C. jejuni* to persist until slaughter age [56].

## 4. Conclusions

In conclusion, this study highlights *C. jejuni*’s ability to establish near-commensal colonization in broilers, transiently altering the gut microbiota’s composition and diversity without significantly compromising microbial function or the broiler’s performance production. Our findings shed light on the host–pathogen interaction, which is an important element for improving poultry health and food safety. However, further studies remain imperative to fully understand the *C. jejuni*–host interaction.

## Figures and Tables

**Figure 1 animals-14-00473-f001:**
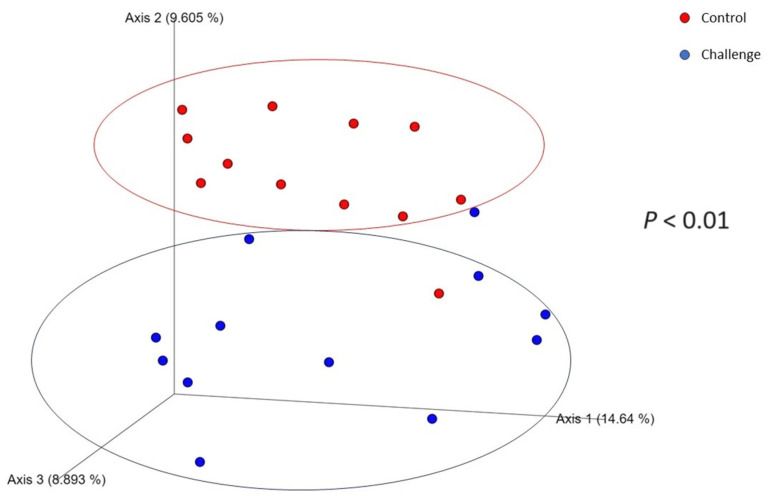
The effect of *C. jejuni* challenge on the Bray–Curtis dissimilarity index. Birds in the challenge group were orally gavaged with 1 × 10^8^ CFU of *C. jejuni*/bird on day 21, and cecal samples were collected on day 28 and day 35 from control (*n* = 6; red) and challenged (*n* = 6; blue) groups. Each dot represents a replicate for each group (*n* = 6) on each day.

**Figure 2 animals-14-00473-f002:**
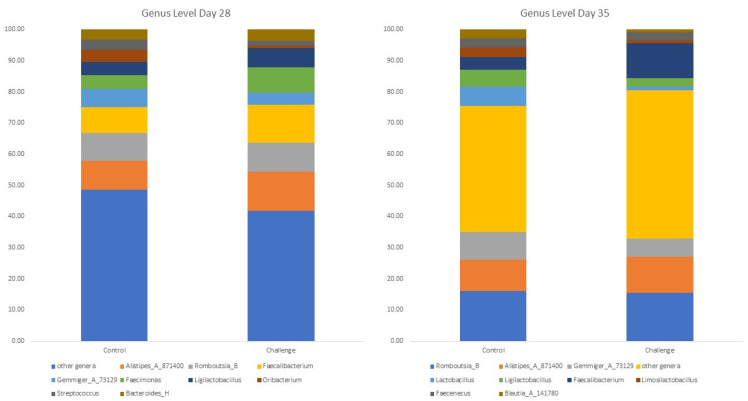
The effect of *C. jejuni* challenge on the top 10 most abundant genera on days 28 and 35. Birds in the challenge group were orally gavaged with 1 × 10^8^ CFU of *C. jejuni/*bird on day 21, and cecal samples were collected on day 28 and day 35 from control (*n* = 6) and challenged (*n* = 6) birds. Following the DNA extraction from the cecal samples, all nine variable regions of the 16S rRNA gene were sequenced (regions V1 to V9). On days 28 and 35, the taxonomic classification was performed using the QIME 2 feature-classifier plugin, which uses the Naïve Bayes classifier trained on the SILVA 138 SSU database.

**Figure 3 animals-14-00473-f003:**
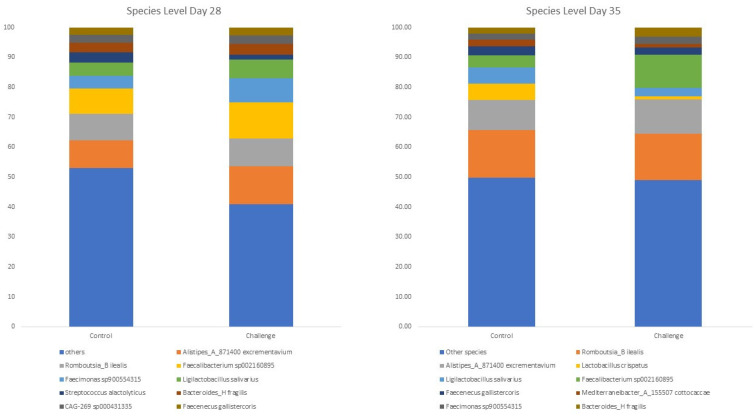
The effect of *C. jejuni* challenge on top 10 species’ relative abundance on days 28 and 35. Birds were orally gavaged with 1 × 10^8^ CFU of *C. jejuni/*bird on day 21, and cecal samples were collected on day 28 and day 35 from control (*n* = 6) and challenged (*n* = 6) birds. Following the DNA extraction from the cecal samples, all nine variable regions of the 16S rRNA gene were sequenced (regions V1 to V9). On day 28 and 35, the taxonomic classification was performed using the QIME 2 feature-classifier plugin, which uses the Naïve Bayes classifier trained on the SILVA 138 SSU database.

**Figure 4 animals-14-00473-f004:**
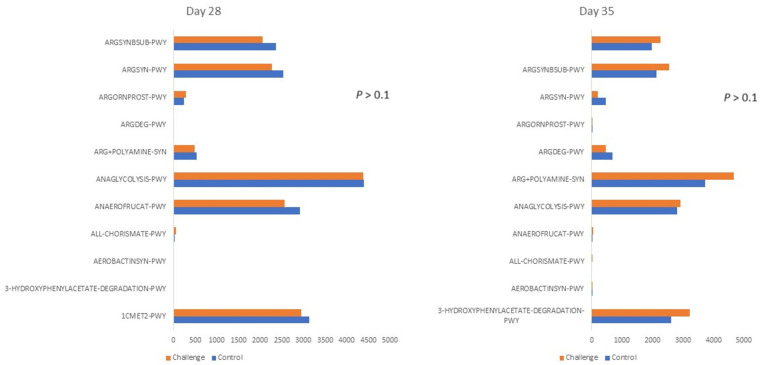
The effect of *C. jejuni* challenge on the microbial function analysis. Birds were orally gavaged with 1 × 10^8^ CFU of *C. jejuni/*bird on day 21, and cecal samples were collected on day 28 and day 35 from control (*n* = 6) and challenged (*n* = 6) birds. On days 28 and 35, the Phylogenetic Investigation of Communities by Reconstruction of Unobserved States (PICRUSt2) was used to make inferences about the metabolic functions of the microbial community, and metagenome metabolic functions were assessed using the MetaCyc pathway database. ARGSYNBSUB-PWY: L-arginine biosynthesis II (acetyl cycle), ARGSYN-PWY: L-arginine biosynthesis I (via L-ornithine), ARGORNPROST-PWY: L-arginine degradation, ARGDEG-PWY: superpathway of L-arginine, putrescine, and 4-aminobutanoate degradation, ARG+POLYAMINE-SYN: superpathway of arginine and polyamine biosynthesis, ANAGLYCOLYSIS-PWY: glycolysis III (from glucose), ANAEROFRUCAT-PWY: homolactic fermentation, ALL-CHORISMATE-PWY: superpathway of horismite metabolism, AEROBACTINSYN-PWY: aerobactin biosynthesis, 3-HYDROXYPHENYLACETATE-DEGRADATION-PWY: 4-hydroxyphenylacetate degradation, 1CMET2-PWY: folate transformations III (E. coli). Mean metabolic pathways indices were analyzed using the Kruskal–Wallis H test.

**Table 1 animals-14-00473-t001:** Basal diet composition fed from day 0 to day 35 of age.

Basal Diet (0–35 d)	%
Corn	58.47
Soybean meal	35.15
Soybean oil	2.269
Limestone	1.593
Biofos	1.387
NaCl	0.35
Vitamins premix ^1^	0.35
Dl-methionine	0.21
Lysine HCL	0.137
Trace mineral premix ^2^	0.08
Total	100
Calculated Nutrient Composition	
ME, kcal/kg	3050
Crude protein, %	21.44
Crude fat, %	4.55
Lysine, %	1.31
Calcium, %	0.95
TSAA, %	0.91
Threonine, %	0.87
Methionine, %	0.56
Available phosphorus, %	0.45

^1^ Vitamin mix per kg of diet: 2.4 mg thiamin-mononitrate, 44 mg nicotinic acid, 4.4 mg riboflavin, 12 mg D-Ca pantothenate, 12 g vitamin B12, 2.7 mg pyridoxine-HCl, 0.11 mg D-biotin, 0.55 mg folic acid, 3.34 mg menadione sodium bisulfate complex, 220 mg choline chloride, 1100 IU cholecalciferol, 2500 IU trans-reinyl acetate, 11 IU all-rac-tocopherol acetate, and 150 mg ethoxyquin. ^2^ Trace mineral mix provided the following per kg diet: 101 mg MnSO_4_.H_2_O, 20 mg FeSO_4_.7H_2_O, 80 mg Zn, 3 mg CuSO_4_.5H_2_O, 0.75 mg ethylene diamine dihydroiodide, 20 mg MgO, and 0.3 mg sodium selenite.

**Table 2 animals-14-00473-t002:** The effect of *C. jejuni* challenge on bird performance.

Body Weight Gain(g)	Control	Challenge	SEM ^1^	*p*-Value
0–21	697	683	12.78	0.28
0–28	1271	1189	33.57	0.11
0–35	1848	1727	54.69	0.14
21–28	574	517	24.79	0.13
21–35	1151	1053	46.10	0.16
28–35	577	537	34.15	0.42
Feed Intake (g)	Control	Challenge	SEM ^1^	*p*-Value
0–21	1331	1267	57.31	0.44
0–28	2244	2075	112.53	0.31
0–35	3257	3053	132.19	0.30
21–28	913	808	74.19	0.34
21–35	1925	1786	92.37	0.31
28–35	1012	977	39.20	0.54
Feed Conversion Ratio	Control	Challenge	SEM ^1^	*p*-Value
0–21	1.90	1.88	0.09	0.89
0–28	1.76	1.75	0.09	0.92
0–35	1.76	1.77	0.08	0.90
21–28	1.6	1.58	0.14	0.93
21–35	1.68	1.70	0.10	0.88
28–35	1.77	1.84	0.10	0.64

Birds in the challenged group were orally gavaged with 1 × 10^8^ CFU of *C. jejuni/*bird on day 21. Average body weight and average feed intake were measured weekly (on days 21, 28 and 35 of age) to calculate the body weight gain and feed conversion ratio (*n* = 6). The two groups were compared using a *t*-test. Different letters in the same row indicate significant differences (*p* ≤ 0.01). ^1^ Standard error of Mean.

**Table 3 animals-14-00473-t003:** Effect of *C. jejuni* on alpha indices.

Alpha Index	Day 28			Day 35		
	Control	Challenge	*p*-Value	Control	Challenge	*p*-Value
Observed features	183	141	0.03	166	148	0.50
Faith Index	9.1	8.7	0.87	10.94	13.24	0.42
Evenness index	0.84	0.81	0.14	0.83	0.81	0.14
Shannon index	6.31	5.79	0.05	6.11	5.82	0.26

Birds in the challenge group were orally gavaged with 1 × 10^8^ CFU of C. jejuni/bird on day 21, and cecal samples were collected on day 28 and day 35 from control (*n* = 6) and challenged (*n* = 6) birds. Following the DNA extraction from the cecal samples, all nine variable regions of the 16S rRNA gene were sequenced (regions V1 to V9). On days 28 and 35, alpha indices were calculated, namely observed features, faith index, Pielou’s evenness index, and Shannon index. Alpha diversity indices were analyzed using the Kruskal–Wallis H test. Different letters in the same row indicate significant differences (*p* ≤ 0.01) and are considered a trend between 0.01 and 0.05.

**Table 4 animals-14-00473-t004:** Effect of *C. jejuni* on the phyla relative abundance.

Phylum (%)	Day 28			Phylum (%)	Day 35		
	Control	Challenge	*p*-Value		Control	Challenge	*p*-Value
*Firmicutes*	86.08	80.68	0.17	*Firmicutes*	86.15	78.42	0.78
*Bacteroidota*	13.43	16.66	0.69	*Bacteroidota*	12.20	14.83	0.93
*Cyanobacteria*	0.21 ^b^	1.73 ^a^	0.01	*Cyanobacteria*	0.26	0.95	0.78
*Patescibacteria*	0.04	0.03	0.73	*Patescibacteria*	0.16	2.99	0.19
*Desulfobacterota*	0.00	0.05	0.40	*Desulfobacterota*	0.00	0.54	0.02
*Proteobacteria*	0.21	0.54	0.92	*Proteobacteria*	0.00	0.36	0.66
*Actinobacteriota*	0.04	0.03	0.78	*Actinobacteriota*	0.05	0.07	0.17
Other phyla	0.00	0.27	0.17	Other phyla	1.17	1.85	0.67

Birds in the challenge group were orally gavaged with 1 × 10^8^ CFU of *C. jejuni/*bird on day 21, and cecal samples were collected on day 28 and day 35 from control (*n* = 6) and challenged (*n* = 6) birds. Following the DNA extraction from the cecal samples, all nine variable regions of the 16S rRNA gene were sequenced (regions V1 to V9). On days 28 and 35, the taxonomic classification was performed using the QIME 2 feature-classifier plugin, which uses the Naïve Bayes classifier trained on the SILVA 138 SSU database. Phyla relative abundance was analyzed using the Kruskal–Wallis H test. Different letters in the same row indicate significant differences (*p* ≤ 0.01) and are considered a trend between 0.01 and 0.05.

**Table 5 animals-14-00473-t005:** Effect of *C. jejuni* challenge on the genera relative abundance. Only genera significantly affected by the challenge are shown.

Genera	Day 28			Genera	Day 35		
	Control	Challenge	*p*-Value		Control	Challenge	*p*-Value
*Acutalibacter*	1.54 ^a^	0.32 ^b^	0.004	*Acutalibacter*	1.08 ^a^	0.34 ^b^	0.01
*UBA644*	0.09 ^a^	0.00 ^b^	0.009	*Oribacterium*	2.45 ^a^	1.63 ^b^	0.01
*Stercorousia*	0.21 ^b^	1.73 ^a^	0.01	*Mailhella*	0.00	0.54	0.02
*UBA1417*	0.49	0.02	0.02	*RUG13038*	0.05	0.42	0.04
*Sellimonas*	0.89	0.24	0.02	*Faecalibacterium*	3.99	11.21	0.04
*Oribacterium*	3.73	0.80	0.04				

Birds in the challenge group were orally gavaged with 1 × 10^8^ CFU of *C. jejuni*/bird on day 21, and cecal samples were collected on day 28 and day 35 from control (*n* = 6) and challenged (*n* = 6) birds. Following the DNA extraction from the cecal samples, all nine variable regions of the 16S rRNA gene were sequenced (regions V1 to V9). On days 28 and 35, the taxonomic classification was performed using the QIME 2 feature-classifier plugin, which uses the Naïve Bayes classifier trained on the SILVA 138 SSU database. Genera’s relative abundance was analyzed using the Kruskal–Wallis H test. Different letters in the same row indicate significant differences (*p* ≤ 0.01) and are considered a trend between 0.01 and 0.05.

**Table 6 animals-14-00473-t006:** Effect of *C. jejuni* challenge on the species relative abundance. Only species significantly affected by the challenge are shown.

Species	Day 28			Species	Day 35		
	Control	Challenge	*p*-Value		Control	Challenge	*p*-Value
*UBA644 sp002299265*	0.09 ^a^	0 ^b^	0.009	*Onthocola_B sp000437355*	0.15 ^b^	0.61 ^a^	0.01
*Sellimonas intestinalis*	0.89	0.24	0.02	*Mailhella massiliensis*	0.00	0.54	0.02
*Evtepia gabavorous*	0.31	0.10	0.03	*Faecalibacterium sp002160895*	3.99	11.17	0.04

Birds in the challenge group were orally gavaged with 1 × 108 CFU of *C. jejuni*/bird on day 21, and cecal samples were collected on day 28 and day 35 from control (*n* = 6) and challenged (*n* = 6) birds. Following the DNA extraction from the cecal samples, all nine variable regions of the 16S rRNA gene were sequenced (regions V1 to V9). On day 28 and 35, the taxonomic classification was per-formed using the QIME 2 feature-classifier plugin, which uses the Naïve Bayes classifier trained on the SILVA 138 SSU database. Species relative abundance was analyzed using the Kruskal–Wallis H test. Different letters in the same row indicate significant differences (*p* ≤ 0.01) and are considered a trend between 0.01 and 0.05.

**Table 7 animals-14-00473-t007:** Short-chain fatty acid concentrations on days 28 and 35.

	Day 28				Day 35			
SCFAs ^1^	Control	Challenge	SEM ^2^	*p*-Value	Control	Challenge	SEM ^2^	*p*-Value
Acetate (mM)	27.58	29.22	2.08	0.59	30.38	33.69	3.14	0.47
Propionate (mM)	1.18	1.15	0.27	0.94	1.20	1.13	0.16	0.76
Butyrate (mM)	3.35	2.27	0.66	0.28	4.70	5.67	0.78	0.40
Isovalerate (mM)	0.02	0.07	0.05	0.44	0.13	0.07	0.04	0.37
Valerate (mM)	0.14	0.17	0.05	0.70	0.28	0.33	0.05	0.51
Total VFA (mM)	32.30	32.91	2.69	0.87	36.72	40.91	3.97	0.47

Birds in the challenge group were orally gavaged with 1 × 10^8^ CFU of *C. jejuni*/bird on day 21, and cecal samples were collected on day 28 and day 35 from control (*n* = 6) and challenged (*n* = 6) birds. On days 28 and 35, short-chain fatty acids’ (SCFAs) concentration was measured using gas chromatograph. SCFAs concentrations were analyzed using *t*-test to determine the difference between the two groups. Different letters in the same row indicate significant differences (*p* ≤ 0.01). ^1^ Short-Chain Fatty Acids, ^2^ Standard Error of Mean.

## Data Availability

The data are available on request from the corresponding author [J.L.].

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
