# Peer review of "Characterizing the Effect of Campylobacter jejuni Challenge on Growth Performance, Cecal Microbiota, and Cecal Short-Chain Fatty Acid Concentrations in Broilers"

_animals, 2024, doi:10.3390/ani14030473_

Round 1

Reviewer 1 Report

Comments and Suggestions for Authors

Comments for animals-2797458 Round 1

In this manuscript, authors tried to investigate the difference between orally gavaged Campylobacter jejuni (C. jejuni) and PBS, found that two genera with opposite numbers, i.e., Sellimonas intestinalis reduction and Faecalibacterium sp002160895 enhancement.

1.      It is first emerging in the main text, the “Campylobacter jejuni”should be abbreviated to the “C. jejuni” in line 44 except the sections of Simple Summary and Abstract.

2.      Note that some symbols should be written in international format, for example the “ml” should be changed to the “mL” in line 31, 90, 108, 116 and 151-154. Besides, the format of table 1-7 should be a three-line grid or magazine requirements. The “4” and “2” were subscripted in the MnSO4.H2O in line 99, including other words or noun FeSO4.7H2O in line 99, CuSO4.5H2O in line 100, B12 in line 96, etc. The “C. jejuni” in line 321 should be italiced

3.      The relevant information on literature statistics is too early. Can authors consider data from the past three years or more recent reports to provide more useful information. Like, the first 8 references (time from 2010 to 2016) are a crucial part of the background content of the entire manuscript, and whether they can represent sufficient information, especially authority, of these contents. This also includes literature from other places.

4.      In the section of results, more references in the initial introduce per paragraph are redundant because it most should transfer to the section of discussion. However, the lacking of section of discussion here according to the “Animals Microsoft Word template file” of instructions for authors of journal Animals.

Comments on the Quality of English Language

The editing of English language required.

Author Response

In this manuscript, authors tried to investigate the difference between orally gavaged Campylobacter jejuni (C. jejuni) and PBS, found that two genera with opposite numbers, i.e., Sellimonas intestinalis reduction and Faecalibacterium sp002160895 enhancement.

  1. It is first emerging in the main text, the “Campylobacter jejuni”should be abbreviated to the “ jejuni” in line 44 except the sections of Simple Summary and Abstract.

Comment:  Fixed

  1. Note that some symbols should be written in international format, for example the “ml” should be changed to the “mL” in line 31, 90, 108, 116 and 151-154.

Comment: Fixed

  1. Besides, the format of table 1-7 should be a three-line grid or magazine requirements.

Comment: Fixed

  1. The “4” and “2” were subscripted in the MnSO4.H2O in line 99, including other words or noun FeSO4.7H2O in line 99, CuSO4.5H2O in line 100, B12 in line 96, etc. The “C. jejuni” in line 321 should be italiced

Comment: fixed

  1. The relevant information on literature statistics is too early. Can authors consider data from the past three years or more recent reports to provide more useful information. Like, the first 8 references (time from 2010 to 2016) are a crucial part of the background content of the entire manuscript, and whether they can represent sufficient information, especially authority, of these contents. This also includes literature from other places.

Comment: Fixed. More recent reports were added in the background content.

  1. In the section of results, more references in the initial introduce per paragraph are redundant because it most should transfer to the section of discussion. However, the lacking of section of discussion here according to the “Animals Microsoft Word template file” of instructions for authors of journal Animals.

Comment: The instruction for authors of journal Animals indicate that the Results and discussion section can be combined together.

Reviewer 2 Report

Comments and Suggestions for Authors

Regarding the manuscript entitled Characterizing the effect of Campylobacter jejuni challenge on growth performance, cecal microbiota, and cecal short-chain 3 fatty acid concentrations in broilers

Abstract, add p-value for significant findings.

L39. This is a strong conclusion please dampen it and your findings may be related to the dose.

L80. Add hypothesis.

L86. What is the breed of broilers?

L90. On which basis did you choose this dose? add ref.

Table 1 is not mentioned within the text. Another critical point on which nutrient guidelines (NRC or breeder standards) the authors fed broilers. 0-35 days, includes three nutritional requirements, starter, grower, and finisher. How did the authors feed the broilers with the same nutritional requirements during the whole period? From 0-35 days is not a starter period.

L170. How did the authors analyse their data by one-way ANOVA? You have only two treatments so it should be a t-test.

Table 2. What is the breed of broilers? The final body weight is lower than the commercial breeds’ final body weight at this age. Another thing what is the sex of these broilers, male or female or mixed sex?

Table 3. Add SEM

Figures 2 and 3. please magnify these figures to be clear for the readers.

Comments on the Quality of English Language

minor editing

Author Response

Regarding the manuscript entitled Characterizing the effect of Campylobacter jejuni challenge on growth performance, cecal microbiota, and cecal short-chain 3 fatty acid concentrations in broilers

  • Abstract, add p-value for significant findings.

Comment: Fixed

  • This is a strong conclusion please dampen it and your findings may be related to the dose.

Comment: Fixed

  • Add hypothesis.

Comment: Fixed

  • What is the breed of broilers?

Comment: Fixed. Cobb 500

  • On which basis did you choose this dose? add ref.

Comment: Fixed. References were added.

  • Table 1 is not mentioned within the text. Another critical point on which nutrient guidelines (NRC or breeder standards) the authors fed broilers. 0-35 days, includes three nutritional requirements, starter, grower, and finisher. How did the authors feed the broilers with the same nutritional requirements during the whole period? From 0-35 days is not a starter period.

Comment: Table 1 shows a basal diet that was fed for the entire experiment. The idea behind feeding a similar diet for the whole experiment is to avoid the effects of changing the feed.

  • How did the authors analyse their data by one-way ANOVA? You have only two treatments so it should be a t-test.

Comment: Fixed. The statistical analysis was carried out using t-test.

  • Table 2. What is the breed of broilers? The final body weight is lower than the commercial breeds’ final body weight at this age. Another thing what is the sex of these broilers, male or female or mixed sex?

Comment: The broiler breed is Cobb 500, and the birds are of mixed sex.

  • Table 3. Add SEM

Comment: The data was analyzed using non-parametric test for non-normal distribution, so SEM is not commonly used in these instances.

  • Figures 2 and 3. please magnify these figures to be clear for the readers.

Comment: Fixed. In order to fit the figures based on the animals template we had to shrink them. The issue is fixed now.

Reviewer 3 Report

Comments and Suggestions for Authors

General comments:  The manuscript is well written, and the authors provide sufficient related literature supporting their findings. The experiment design and analysis performed were straightforward and appropriate. Moreover, the authors also give relevant limitations of their work and timely recommendations to further the science in understanding C. jejuni infection in broilers.

Related comments are in the attached file.

Author Response

Title: Characterizing the effect of Campylobacter jejuni challenge on growth performance, cecal

microbiota, and cecal short-chain fatty acid concentrations in broilers.

Brief Summary: The manuscript presents data on the effects of C. jejuni in broilers, highlighting

the changes in the microbial composition of the ceca and its related microbial functional

pathways. Their findings added insight into the context of host-pathogen interaction following

microbiome analysis.

General comments: The manuscript is well written, and the authors provide sufficient related

literature supporting their findings. The experiment design and analysis performed were

straightforward and appropriate. Moreover, the authors also give relevant limitations of their

work and timely recommendations to further the science in understanding C. jejuni infection in

broilers.

Specific comments:

  1. Table footnotes: Please write appropriate footnotes on all the tables indicating necessary

elements that apply presented in the table like superscripts, significance (p.values),

acronyms, and how data are presented (i.e., average,)

Comment: Fixed

  1. Materials and methods: Please include the type/breed/strain of broiler chickens used in

the experiment.

Comment: Fixed

  1. Figure descriptions: Please improve the description of all figures detailing the followinga. A declarative title, followed briefly by methods used, how data are presented,when appropriate, the statistical information and interpretation.
  2. Example
  3. Fig.1. Beta diversity plot of the C. jejuni challenged broilers. Birds were orally gavaged with 1x108 CFU of C. jejuni on day 21, and cecal samples were collected on day 28 and day 35 from control (N=6; red) and challenged (N=6; blue)groups. Each dot represents a replicate for each group. The beta diversity, the more similar the soil microbial communities of the samples, were calculated using Bray-Curtis dissimilarity.

Comment: Fixed

  1. Include Table 4 in the main text (Section 3.3. Bacterial diversity at the phylum level)

Comment: Fixed

  1. Lines 248-252: Please include the reference.

Comment: Fixed

  1. Lines 272: Table 6 presents the relative abundance of species. Hence the sentence

needs to be revised to "different species" from 'different genera'.

Comment: Fixed

  1. Figure 4, Day 35: Missing label on the axis. Please make sure all necessary labels and

legends are presented in the figure.

Comment: Fixed

  1. Others
  2. Lines 59, 60-61: Consider using "avian intestinal mucus" instead of 'avian mucus'

Comment: Fixed

  1. Line 88: 'ad libtium' to "ad libitum"

Comment: Fixed

  1. Line 209, 211: 'Alfa' to "Alpha"

Comment: Fixed

Reviewer 4 Report

Comments and Suggestions for Authors

Table #1 the Calculated ingredients should be placed under the feed ingredients. 

Table #2 There seems to be a difference in body weight from the challenge birds compared to the control at 0-35 days. Same for your feed intake 0-28 days and 21-25 days. 

Figure 4 seems to show a difference in microbial activity at day 35.

Comments on the Quality of English Language

English is average.

Author Response

  • Table #1 the Calculated ingredients should be placed under the feed ingredients. 

Comment: Fixed.

  • Table #2 There seems to be a difference in body weight from the challenge birds compared to the control at 0-35 days. Same for your feed intake 0-28 days and 21-25 days. 

Comment: We carried a t-test to analyze the difference between the birds performance for all the measured weeks. There is a numerical difference however, that was not indicated in the P-value.

  • Figure 4 seems to show a difference in microbial activity at day 35.

Comment: We used Kruskal Wallis to analyze the microbial activity at day 28 and 35. We designated P ≤0.01 as the significance, and any P-value between 0.01 and 0.05 as trend effect. All the microbial activates didn’t fall within the designated P-value, and this is indicated on the figure. 

Round 2

Reviewer 1 Report

Comments and Suggestions for Authors

Comments for animals-2797458 Round 2

The authors made a carefully modified in the peer revision-v2 manuscript, and further changes have been made according to the magazine's requirements. Please note that the “P” valule should be italicized in manuscript.

Author Response

Comments for animals-2797458 Round 2

The authors made a carefully modified in the peer revision-v2 manuscript, and further changes have been made according to the magazine's requirements. Please note that the “P” valule should be italicized in manuscript

Comment: Fixed. The “P” values are now italicized.

Reviewer 2 Report

Comments and Suggestions for Authors

Thank you for your revisions.

Comments on the Quality of English Language

minor editing

Author Response

Thank you for your comments, as they have helped us to improve our manuscript significantly.

Reviewer 4 Report

Comments and Suggestions for Authors

Please indicate if the microbial activity was actually changed or not. 

Author Response

Please indicate if the microbial activity was actually changed or not. 

Comment: Fixed. In lines 324-327, it is indicated that the challenge didn’t affect the metabolic pathways. Also, the picture is now fixed, and it shows P > 0.1, so we don’t have a trend as well in our P-values. It is non-significant.